# Monascuspiloin from *Monascus*-Fermented Red Mold Rice Alleviates Alcoholic Liver Injury and Modulates Intestinal Microbiota

**DOI:** 10.3390/foods11193048

**Published:** 2022-09-30

**Authors:** Li Wu, Kangxi Zhou, Ziyi Yang, Jiayi Li, Guimei Chen, Qi Wu, Xucong Lv, Wenlin Hu, Pingfan Rao, Lianzhong Ai, Li Ni

**Affiliations:** 1College of Chemistry, Fuzhou University, Fuzhou 350108, China; 2Research Institute of Agri-Engineering and Technology, Fujian Academy of Agricultural Sciences, Fuzhou 350003, China; 3Institute of Food Science and Technology, College of Biological Science and Technology, Fuzhou University, Fuzhou 350108, China; 4Food Nutrition and Health Research Center, School of Advanced Manufacturing, Fuzhou University, Jinjiang 362200, China; 5Guangdong Tianyi Biotechnology Co., Ltd., Zhanjiang 524300, China; 6Shanghai Engineering Research Center of Food Microbiology, School of Medical Instrument and Food Engineering, University of Shanghai for Science and Technology, Shanghai 200093, China

**Keywords:** *Monascus*, monascuspiloin, alcoholic liver injury, intestinal microbiome, liver metabolomics

## Abstract

*Monascus*-fermented red mold rice (RMR) has excellent physiological efficacy on lipid metabolism and liver function. This study investigated the ameliorative effects of monascuspiloin (MP) from RMR on alcoholic liver injury in mice, and further clarified its mechanism of action. Results showed that MP intervention obviously ameliorated lipid metabolism and liver function in mice with over-drinking. In addition, dietary MP intervention reduced liver MDA levels and increased liver CAT, SOD, and GSH levels, thus alleviating liver oxidative stress induced by excessive drinking. 16S rRNA amplicon sequencing showed that MP intervention was beneficial to ameliorate intestinal microbiota dysbiosis by elevating the proportion of norank_f_*Lachnospiraceae*, *Lachnoclostridium*, *Alistipes*, *Roseburia*, *Vagococcus*, etc., but decreasing the proportion of *Staphylococcus*, norank_f_*Desulfovibrionaceae*, *Lachnospiraceae*_UCG-001, *Helicobacter*, norank_f_*Muribaculaceae*, unclassified_f_*Ruminococcaceae*, etc. Additionally, correlation network analysis indicated that the key intestinal bacterial taxa intervened by MP were closely related to some biochemical parameters of lipid metabolism, liver function, and oxidative stress. Moreover, liver metabolomics analysis revealed that dietary MP supplementation significantly regulated the levels of 75 metabolites in the liver, which were involved in the synthesis and degradation of ketone bodies, taurine, and hypotaurine metabolism, and other metabolic pathways. Furthermore, dietary MP intervention regulated gene transcription and protein expression associated with hepatic lipid metabolism and oxidative stress. In short, these findings suggest that MP mitigates alcohol-induced liver injury by regulating the intestinal microbiome and liver metabolic pathway, and thus can serve as a functional component to prevent liver disease.

## 1. Introduction

Excessive alcohol consumption can lead to liver metabolic disorder, oxidative stress, and inflammatory response, promoting the development of alcoholic liver disease (ALD), which is also the leading cause of chronic liver disease worldwide [1]. Mounting evidence demonstrates that excessive drinking would produce a large number of toxic acetaldehyde and reactive oxygen species in the liver, which are difficult to clear in time, resulting in liver injury, which may progress to steatosis, steatohepatitis, fibrosis, cirrhosis, and even cancer [2]. Although a large number of therapeutic drugs have been developed for the treatment of alcoholic liver injury, some of them have certain side effects. The drug disulfiram, approved by the Food and Drug Administration (FDA) of the United States in 1983 to treat alcoholic liver disease, has little evidence that it enhances abstinence, is poorly tolerated, and is now rarely used. Naltrexone, approved in 1995 to treat alcoholism, controls alcohol cravings but can also cause liver cell damage [3]. Therefore, the development of natural hepatoprotective products is an effective strategy to prevent the occurrence of ALD.

As one of the major secondary metabolites of *Monascus* spp. [4], *Monascus* azaphilone pigments (MonAzPs) have been widely used as an important food colorant in daily life for thousands of years in Southeast Asia. Moreover, MonAzPs have received more and more attention in recent years due to their various physiological activities. Until now, more than one hundred MonAzPs have been successfully identified from *Monascus*-fermented red mold rice (RMR, also called *Hongqu*), and the metabolites of *Monascus* spp. [4], most of which possess significant pharmacological activities, including hypolipidemic, hypoglycemic, and anti-aging properties, etc. The beneficial effects of MonAzPs, monacolin K, ergosterol, polysaccharides, and γ-aminobutyric acid on the antioxidant activity of RMR were previously investigated through in vitro assay, and the results revealed that the content of MonAzPs was positively correlated with the antioxidant capacity [5]. Further investigation on MonAzPs components revealed significant differences in the composition of *Monascus* yellow pigment (MYPs) between the functional *Hongqu* and the coloring *Hongqu*, and five MYPs purified from coloring *Hongqu* have strong cellular antioxidant activity [5]. As a typical representative of MYPs in RMR, monascin (MS) was proven to modulate ethanol-induced PPAR-γ and SREBP-1 expressions in HepG2 cells [6]. Besides, MS activated the expression of PPAR-γ protein while enhancing anti-inflammatory and antioxidant pathways to prevent liver damage in the ALD mice [7]. The literature reported that monascuspiloin (MP, an analogue of MS) has therapeutic potential for the treatment of both androgen-dependent and -independent human prostate cancers [8,9]. Moreover, MP can treat hereditary alopecia and prostatic hypertrophy by inhibiting the activity of pentamethyl reductase and interfering with the binding of dihydrotestosterone to its receptor. Alcohol metabolism inevitably leads to abnormal changes in liver metabolite levels and metabolic pathways. However, the effects of MP intervention on intestinal microflora disorder and liver metabolic disorder caused by excessive drinking and its possible molecular mechanism have not been reported.

It is commonly accepted that the abnormality of the gut microbiome plays a crucial role in the pathogenesis of alcohol-induced liver damage [10]. Alcoholism may cause the disturbance of the intestinal flora, then damage the intestinal barrier and promote the circulation of pathogenic metabolites into the liver, thereby resulting in damage to liver function [11,12]. Over-drinking markedly altered the feature of intestinal microbiota and microbiota-derived metabolites [13,14]. Significant increases in the abundance of beneficial bacteria in the gastrointestinal tract, including *Lactobacillus*, *Bifidobacterium,* and *Akkermansia*, are known to improve intestinal barrier and liver metabolic capacity [15]. Conversely, some harmful metabolites produced by specific intestinal bacterial taxa, such as lipopolysaccharides produced by *Klebsiella*, *Veronaceae,* and *Streptococcaceae*, may go through the intestinal barrier and enter the liver or serum to cause severe inflammatory response [16]. A large amount of research evidence has shown that alcohol-induced liver damage is significantly correlated with changes in intestinal flora composition. Hence, specific modulation of the intestinal microbial composition through dietary interventions would be one of the effective ways to ameliorate alcoholic liver injury [15]. According to untargeted liver metabolomics analysis, the FXR/RXR activation pathway, which is closely related to bile acid synthesis, was most significantly up-regulated in ALD animals [17]. Alcohol-regulated liver metabolite levels were restored after treatment with ginger essential oil [18]. Therefore, liver metabolomics analysis is helpful to comprehensively understand the potential pathway of MYP for the prevention of ALD and elucidate the molecular mechanism of its hepatoprotective effect. There is a close correlation between diet and intestinal microbiome, but MP intervention affects the intestinal microbial composition, and liver metabolic function is still unknown. The modulation of the intestinal microbiome and liver metabolic might be partially responsible for the ameliorative effects of MP on alcoholic liver injury.

In this study, we elucidated the ameliorative effects and possible mechanism of action of MP intervention against alcoholic liver injury from the perspective of intestinal microbiomics and liver metabolomics, coupled with liver gene transcription and protein expression analysis. Moreover, the associations of the intestinal microbiota features with the biochemical parameters were demonstrated through correlation analysis, providing a scientific reference for the development of special diets or medicines for people with excessive alcohol consumption.

## 2. Materials and Methods

### 2.1. Materials and Chemical Reagents

MP and MS monomer compounds were prepared from RMR by means of 70% ethanol extraction, column chromatography, and thin-layer chromatography [9,19]. The purity of MP and MS (greater than 95%) was confirmed through the HPLC-QTOF-MS system (Agilent HPLC1260-MS6530, Santa Clara, CA, USA) (Appendix A). Detection kits for serum and liver biochemical parameters were obtained from Elabscience Biotechnology Co., Ltd. (Wuhan, Hubei, China). The primary antibodies of low-density lipoprotein receptor (LDLr) (#ab52818), carnitine palmitoyltransferase-1 (CPT-1) (#ab234111), quinone oxidoreductase 1 (NQO1) (#ab 80588), nuclear factor erythroid 2-related factor 2 (Nrf-2) (#ab92946), heme oxygenase-1 (HO-1) (#ab52947), and SREBP-1 (#ab105136) for Western Blotting (WB) were obtained from Abcam Co., Ltd. (Cambridge, UK). β-actin (#YT0099) and HRP-conjugated anti-rabbit secondary antibody (#RS0002) were purchased from Ruiying Biotech. Co., Ltd. (Jiangsu, China). Unless otherwise specified, other chemicals were provided by Shanghai Sangon Biotech Co., Ltd. (Shanghai, China).

### 2.2. Animal Experiment

Male Kunming mice without specific pathogens (*n* = 34, six-week-old) were obtained from Jinfeng Experimental Animal Co., Ltd. (Jinan, Shandong, China). All mice were kept in a laboratory of specific pathogen-free (SPF) grade (temperature 22 ± 2 °C, humidity 58 ± 2%, 12-h/12-h light/dark cycle), and sterile diets and sterilized tap water were provided ad libitum to the mice during the experiment. The dose settings of monascuspiloin (MP) and monascin (MS) in this study were mainly based on the daily consumption of Monascus-fermented red mold rice recommended by the Chinese Nutrition Society and relevant literature reports. Based on the experimental dosage of MS in many previous studies on alcohol-induced injury [7,20,21], anti-inflammatory [22], and hypoglycemic [23], the MS dose in our trial was determined to be 10 mg/kg b.w./day. Taking MS as a positive control, the protective effect of MP on the alcoholic liver injury was investigated at the same dosage of 10 mg/kg b.w./day to conduct a comparative study. After a 7-days adaptation period, mice were randomized as follows: the control group (*n* = 8), the model group (*n* = 10), the MS group (*n* = 8, MS, 10 mg/kg b.w./day) and the MP group (*n* = 8, MP, 10 mg/kg b.w./day). The mice in the control group were gavaged with physiological saline; the mice in the model group were gavaged with 50% ethanol (*v*/*v*, 7.5 mL/kg b.w./day); the mice in the MS and MP groups were gavaged with MS and MP suspension, respectively, supplemented with 50% ethanol (*v*/*v*, 7.5 mL/kg b.w./day).

After six weeks of experimental feeding, all mice were fasted overnight and then anesthetized with ketamine solution (100 mg/kg b.w.) according to the recommendations for anesthesia of experimental animals. Immediately after a laparotomy, blood was collected from the vena cava into tubes and stored at 20 °C for 0.5 h, and then centrifuged at 500 g and 8 °C for 20 min to obtain the serum. Fresh cecal contents were obtained and placed in 2.0 mL sterile tubes. The fresh livers were snap-frozen in liquid nitrogen and stored at −80 °C until testing. The experiment was approved by the Laboratory Animal Ethics Committee of the first author’s workplace to ensure the welfare of the animals (FZU-FST-2021-016).

### 2.3. Biochemical Parameter Detection

Biochemical parameters were determined with an automatic biochemical analyzer (Toshiba TBA-40FR, Tokyo, Japan). According to the kit instructions, a high-speed homogenizer was used to prepare a 10% liver tissue homogenate for the detection of the liver biochemical index. Fecal samples were pretreated in the same way as liver samples, and the supernatants were prepared for the quantitative detection of fecal TC, TG, and BAs.

### 2.4. Histological Examination

Histological sections of the liver, jejunum, and colon were prepared according to the literature and used for histopathological analysis [12]. After hematoxylin-eosin (H&E) staining, tissue sections were observed with a light microscope and photographed, respectively (Nikon, Tokyo, Japan).

### 2.5. Analysis of Fecal Short-Chain Fatty Acids (SCFAs) by Gas Chromatograph (GC)

Fecal samples were obtained according to the method from the standard (http://www.microbiome-standards.org accessed on 1 January 2015). According to the previous report by Guo et al. [24], the extraction and determination of SCFAs in feces were carried out. Agilent 7890B GC system (Agilent, Santa Clara, CA, USA) was used for component separation and detection of the fecal SCFAs concentration.

### 2.6. Amplicon Sequencing and Bioinformatics Analysis

Total bacterial DNA of cecal content was extracted with a fecal DNA extraction kit (MoBio, San Mateo, CA, USA), and then amplified the 16S rDNA V3-V4 sequencing with 341F-806R primers based on the Illumina MiSeq platform of Majorbio Co., Ltd. (Shanghai, China) [19]. Hierarchical clustering analysis was carried out and visualized by SIMCA-14.1 software (UMETRICS, Umea, Sweden). Differential OTUs abundance analyses based on negative binomial distribution were performed using STAMP software (Ver. 2.1.3, Parks DH, Tyson GW, Hugenholtz P, Beiko RG., Halifax, Canada). The correlations between the key bacterial OTUs with biochemical parameters were revealed by Spearman’s correlation analysis, visualized by heatmap and network through R software (Ver. 3.3.3, Becker R, Chambers J, Wilks A, New York, NY, USA) and Cytoscape software (Ver. 3.9.0, Institute for Systems Biology, Seattle, WA, USA), respectively. The sequence data reported in this study were archived in the Sequence Read Archive (SRA) with the accession number PRJNA872915.

### 2.7. Metabolomics Analysis

Liver samples were extracted according to the method in the previously reported literature [25]. The extracts of liver tissue were analyzed using a UPLC-QTOF/MS system (Agilent 1290 Infinity UPLC equipped with Agilent 6530 MS). Assay conditions are detailed in the previously published literature [26]. SIMCA 15.0 software was used to perform multivariate statistical analysis. OPLS-DA was used to screen the liver metabolites with significant differences between the model and MP groups (VIP value > 1.0, *p* < 0.05). Finally, MetaboAnalyst 5.0. the online platform was used for pathway analysis.

### 2.8. RT-qPCR

Total RNA was extracted from the liver and reverse transcribed into cDNA according to commercial kits (Takara, Beijing, China). qPCR amplification was performed using QuantStudio 7 Pro Real-Time qPCR System (Thermo Scientific, Waltham, MA, USA). The expression levels of selected genes were normalized with the 18S rRNA gene. The qPCR data were analyzed according to the 2^−ΔΔCT^ standard method. Table 1 shows the primers used in the current study.

### 2.9. Western Blot

Total liver proteins were extracted with RIPA lysate containing protease inhibitors, and the protein concentration was quantified by a commercial quantification kit (Beyotime, Beijing, China). Protein solutions of appropriate concentration were loaded to SDS-PAGE gel and then transferred to polyvinylidene fluoride membranes. Afterward, the membranes were blocked with 5% defatted milk in Tris-buffered saline with Tween 20 (pH 8.0), and further incubated at 4 °C overnight with the primary antibodies (1:1000), incubated for an additional 2 h with the secondary antibody (1:20,000). Protein bands were determined using the GBOX-CHE MIXRQ Chemiluminescence & Fluorescence Systems (Beijing, China). The optical density was quantitated using Carestream MISE V5.3.5 software (Carestream Health, Inc., Toronto, Canada).

### 2.10. Statistical Analysis

All data of this study are indicated as the mean ± standard deviation. The statistical significance was analyzed by one-way analysis of variance with GraphPad Prism (ver 8.0, GraphPad Software, San Diego, CA, USA). Data with different superscripts are significantly different from each other at *p <* 0.05. ^##^ *p* < 0.01 and ^#^ *p <* 0.05, versus the Control group; ** *p* < 0.01 and * *p* < 0.05, versus the Model group.

## 3. Results

### 3.1. Effects of MP Intervention on the Body Weight and Organ Indexes

Before the intervention, there was no significant difference in the body weight of mice in each experimental group. The weight of mice given 50% ethanol by gavage for 6 weeks decreased significantly (*p* < 0.05), indicating that excessive drinking has adverse effects on body metabolism (*p* < 0.05) (Figure 1). As reported, excessive drinking may damage the growth and metabolism of mice, thus affecting weight gain [26]. MS and MP intervention (10 mg/kg b.w.) reversed the alcohol-induced abnormal reduction of body weight. Compared with the control group, excessive drinking remarkably increased the liver index (*p* < 0.01). Interestingly, MS and MP intervention (10 mg/kg b.w.) decreased the liver index with inhibition rates of 17.17 % and 13.01 %, respectively. However, in this study, both EtOH treatment, MP, and MS intervention had no significant effects on the spleen index and kidney index.

### 3.2. Effects of MP on Serum Biochemical Parameters

Lower levels of serum TC and TG are beneficial for reducing the risk of atherosclerosis and liver injury [10]. It was previously reported that serum LDL-C level was closely related to the incidence of hyperglycemia and hyperlipidemia [27]. Compared with mice without alcohol treatment, the serum levels of TC, TG, and LDL-C in mice of the model group were significantly increased, and HDL-C was significantly decreased (*p* < 0.01), indicating that the alcoholic liver injury model was successfully established (Figure 2). The serum TC, TG, and LDL-C levels of the MP-treated ALD mice were significantly decreased, while the HDL-C level was significantly increased (*p* < 0.05). Moreover, serum ALT and AST activities were increased distinctly in ALD mice (*p* < 0.01). Interestingly, serum ALT and AST activities were significantly decreased by MP treatment, reflecting that the liver function was alleviated [7]. In particular, the effect of MP treatment on serum biochemical indexes of mice with excessive drinking was better than those of mice supplemented with the same dose of MS.

### 3.3. Effects of MP on Liver Biochemical Parameters and Histopathological Features

As reported, alcoholism causes abnormal accumulation of lipids and bile acids (BAs), resulting in liver metabolic dysfunction and liver injury [28]. The hepatic TC and BAs levels were abnormally elevated in over-drinking mice compared with healthy mice, (Figure 3A). The abnormal accumulation of BAs in the liver was obviously inhibited in MP or MS-treated mice. Oxidative stress status is another important index reflecting the degree of liver damage caused by excessive drinking. Compared with the control group, hepatic GSH level, hepatic CAT, and SOD activities of the model group were decreased distinctly, but MDA content and LDH activity were elevated markedly (*p* < 0.01). It is well known that GSH with active sulfhydryl groups susceptible to oxidative dehydrogenation readily scavenges free radicals [29]. MDA is an important indicator of oxidative stress. MP or MS supplementation distinctly increased the hepatic GSH content and activity of antioxidant enzymes (CAT), but significantly decreased the hepatic MDA level, suggesting that MP or MS intervention reducing alcoholic liver injury may be partly achieved by ameliorating oxidative stress. In addition, the MP intervention obviously increased the SOD activity, while that of MS was not significantly different. ADH catalyzes the EtOH breakdown to produce acetaldehyde in the liver. The results showed that MP or MS supplementation increased the activity of ADH in the liver of mice with over-drinking. The effect of MP in enhancing LDH enzyme activities was better than that of MS.

Furthermore, histopathological examination of the liver with H&E staining (Figure 3B) showed that the normal mice of the control group had clear hepatic lobular structure, and radioactive arrangement of liver cords. However, mice with over-drinking had an abnormal accumulation of lipids in hepatocytes. The degree of steatosis of hepatocytes in MP-treated mice was significantly improved than in the ALD mice, which was similar to the microscopic morphology of liver tissue in the control group.

### 3.4. Effects of MP on Fecal Biochemical Parameters and Intestinal Histopathological Features

Excretion of cholesterol, triglycerides, and bile acids in feces is an important way to reduce the excessive lipid accumulation in the liver. As a product of cholesterol degradation, bile acids can be excreted with food residues in the intestine and play an important role in fat metabolism. SCFAs are generally produced by anaerobic fermentation of non-digestible carbohydrates and have beneficial effects on host health, mostly due to the modulation of metabolism, inflammation, lipogenesis, and intestinal immunity [30]. The fecal SCFAs content was reduced in over-drinking mice than in healthy mice (Figure 4A), which was consistent with the previous studies [31,32]. It is worth noting that 10 mg/kg b.w. of MP or MS intervention obviously enhanced the fecal contents of acetate, propionate, isobutyrate, butyrate, isovalerate, and valerate in mice with excessive drinking (*p* < 0.05). The effect of MP in enhancing the levels of fecal butyrate was better than that of MS. Among them, acetic acid is mainly involved in the energy metabolism of the body, which is one of the main ways for the human body to obtain energy from carbohydrates that cannot be absorbed [33]. Propionic acid is also widely used as energy, which can inhibit cholesterol synthesis in the liver, so it can effectively ameliorate cholesterol metabolism disorder caused by ethanol toxicity [34]. It is noteworthy that butyric acid provides energy for colon and cecum epithelial cells, but also maintains the integrity of the intestinal barrier and intestinal homeostasis [35]. MP and MS intervention may ameliorate liver damage by increasing the fecal levels of SCFAs produced by gut microbes. Therefore, the beneficial effects of MP and MS on the gut microbiota are particularly important.

This study also investigated the histopathological changes in the structure of the colon and jejunum of mice. In the control group, the jejunum had clear layers, the mucosal epithelium and glands were structurally intact, and the epithelial cells were closely connected with the lamina propria (Figure 4B) [28]. Compared with the control group, the colonic mucosa epithelial cells of the mice with excessive drinking were sloughed off, the distance between the epithelial cells and the lamina propria was significantly widened, the glandular structure was destroyed, and the crypts became shallower. MP and MS intervention significantly reverse the damage of jejunum mucosa caused by excessive drinking. In the control group, the colonic mucosa epithelial cells were large and numerous, the villi were arranged closely and the crypt was deep (Figure 4C) [36]. Excessive alcohol consumption resulted in smaller and fewer epithelial cells, wider villi, shallower crypts, thinner and lighter mucosa, submucosa, and musculus. Notably, MP and MS intervention significantly improved the degree of damage to colonic mucosa caused by excessive alcohol consumption. The effect of MP in maintaining the structure and function of colonic mucosal was better than MS, which was closely related to the function of butyrate up-regulated by MP.

### 3.5. Effects of MP Intervention on the Composition of Intestinal Microflora

The overall changes in intestinal microbiota were analyzed by PCoA and hierarchical cluster. The PCoA plot illustrated that the model mice was obviously distinguished from mice of the control group in the direction of the second principal axis (PCoA [2]) (Figure 5A), indicating that excessive drinking could induce the composition disorder of intestinal microbiota. The MP group was clearly separated from the model group in the direction of the first principal axis (PCoA [1]) (Figure 5A), suggesting that the intestinal microbiota composition was evidently regulated by MP intervention (10 mg/kg b.w.). Hierarchical clustering based on the relative abundance of OTUs also indicated that MP intervention obviously influenced the composition of intestinal microflora in mice with excessive drinking (Figure 5B).

Differential analysis of the relative abundance of OTUs between different two groups revealed the effects of excessive drinking and MP intervention on the composition of intestinal microflora. The model group exhibited elevated proportions of eight intestinal bacterial taxa (such as *Lachnospiraceae* NK4A136 group (OTU375, 632, 374, and 326), etc.), but lowered the relative abundances of sixteen intestinal bacterial taxa (such as *Lactococcus* (OTU115), etc.) as compared with the control group, indicating that intestinal bacterial dysbiosis occurred in alcohol-impaired mice (Figure 6A). Correlation analysis has shown that there was a significant correlation between the incidence of ALD and the abundance of *Lachnospiraceae* and *Muribaculaceae* [37,38]. The comparison between the model and MP groups showed that dietary supplementation with 10 mg/kg b.w. MP remarkably increased the relative abundance of unclassified_o_*Oscillospirales* (OTU778), *Coriobacteriaceae*_UCG-002 (OTU629, 272), *Ruminococcus* (OTU555), and norank_f_*Peptococcaceae* (OTU653), but reduced the proportion of unclassified_f_*Ruminococcaceae* (OTU150), norank_f_*Lachnospiraceae* (OTU347, 304), norank_f_*Desulfovibrionaceae* (OTU494), *Oscillibacter* (OTU756), *Lachnospiraceae*_UCG-001 (OTU615), *Butyricicoccus* (OTU128), norank_f_norank_o_*Clostridia*_vadinBB60_group (OTU552), norank_f_*Ruminococcaceae* (OTU853), norank_f_*Oscillospiraceae* (OTU837), unclassified_f_*scillospiraceae* (OTU628), *Lachnospiraceae*_NK4A136_group (OTU632), UCG-005 (OTU159), unclassified_c_*Clostridia* (OTU845), *Blautia* (OTU291), unclassified_f_*Lachnospiraceae* (OTU366, 858), *Ruminococcus* (OTU62), *Exiguobacterium* (OTU195), and *Anaerotruncus* (OTU116) (Figure 6B). As the common members of the intestine, *Oscillospirales* were the main producers of butyric acid [39]. Butyrate is an important reference index for screening “next-generation probiotics”. So *Oscillospirales* is also listed as a candidate for next-generation probiotics [40]. The order *Coriobacteriales* usually has a strong glycolytic ability and can metabolize a variety of carbohydrates to produce metabolites such as lactic acid [41]. Besides, *Coriobacteriales* was positively correlated with acetic acid and negatively correlated with inducible nitric oxide synthase (iNOs) activity [42]. *Grifola frondosa* ethanolic extract increases the abundance of *Ruminococcus*, and modulates the functional composition of the gut microbiota to ameliorate glycolipid disorders [43]. There was a positive correlation between Peptococcaceae and the mRNA expression of dopamine receptor 1 (D1R), dopamine receptor 2 (D2R), tyrosine hydroxylase (TH), and then increased the intake of high sugar and high-fat diet during obesity [44]. In contrast, unclassified *Clostridiales*, *Butyricicoccus*, and *unclassified Lachnospiraceae* were positively associated with alcoholic liver disease [45]. *Desulfovibrionaceae* in the intestine can metabolize choline into trimethylamine (TMA), which is further transformed into trimethylamine-N-oxide (TMAO), harmful to human health by heparin monooxygenase, leading to the development of hyperlipidemia and fatty liver [46].

### 3.6. Correlations of the Key Bacterial OTUs with the Biochemical Parameters

The mechanisms of gut microbes regulating the physiological activities of the host’s gut, immune and nervous systems, and signaling by secreting SCFAs are impressive. Therefore, the correlations between key gut bacterial OTUs and biochemical parameters significantly modulated by MP intervention were elucidated by Spearman’s correlation analysis and visualization network. Serum HDL-C, fecal TC, fecal TBA, hepatic CAT, hepatic SOD, hepatic GSH, hepatic ADH, and fecal SCFAs were good indicators. Serum TC, TG, LDL-C, ALT, AST, and hepatic MDA were important indicators for alcoholic liver injury. As shown in Figure 7A,B, unclassified_o_*Oscillospirales* (OTU778) showed a negative correlation with hepatic ADH and GSH, but a positive correlation with serum TG and fecal TG. In addition, *Coriobacteriaceae*_UCG-002 (OTU629, 272) were positively correlated with the hepatic ADH level, but negatively associated with fecal valerate. It is clear that *Ruminococcus* (OTU555) correlated negatively with the serum LDL-C level. Norank_f_*Peptococcaceae* (OTU653) was negatively correlated with serum TG levels, serum AST and ALT activities. These results suggest that unclassified_o_*Oscillospirales* (OTU778), *Coriobacteriaceae*_UCG-002 (OTU629, 272), *Ruminococcus* (OTU555), and norank_f_*Peptococcaceae* (OTU653) were beneficial bacteria for the prevention of alcoholic liver damage. These findings suggest that the prevention of alcoholic liver damage by MP intervention may be partly related to the regulation of intestinal microbial composition. The MP-impoverished norank_f_norank_o_*Clostridia*_vadinBB60_group (OTU552) exhibited positive correlations with the serum AST and fecal valerate, but showed negative relationships with the hepatic ADH and CAT. Moreover, unclassified_f_*Ruminococcaceae* (OTU150) and *Ruminococcus* (OTU62) were negatively correlated with hepatic ADH. norank_f_*Ruminococcaceae* (OTU853) was found to be negatively related to hepatic ADH, but positively correlated with hepatic MDA. norank_f_*Lachnospiraceae* (OTU304, 347) showed a positive correlation with serum TC level, but a negative correlation with hepatic ADH and fecal propionate. *Blautia* (OTU291) and *Anaerotruncus* (OTU116) were found to be positively related to serum TG, but negatively associated with hepatic ADH and fecal propionate. The *Lachnospiraceae*_UCG-001 (OTU65) were positively correlated with fecal TG, but negatively associated with fecal TBA. It was clear that *norank_f_Oscillospiraceae* (OTU837) and *Oscillibacter* (OTU756) correlated positively with the hepatic MDA, were negatively correlated with hepatic ADH, fecal propionate, acetate, and isobutyrate levels. *Butyricicoccus* (OTU128) exhibited positive correlations with the serum TG, but showed negative relationships with the hepatic ADH and LDH. There were negative relationships between *unclassified_f_Lachnospiraceae* (OTU366) with the fecal propionate. Simultaneously, *Exiguobacterium* (OTU195) was negatively associated with hepatic ADH and CAT, and *norank_f_Desulfovibrionaceae* (OTU494) was negatively associated with hepatic_ADH and fecal propionate. These results suggested that norank_f_norank_o_*Clostridia*_vadinBB60_group (OTU552), unclassified_f_*Ruminococcaceae* (OTU150), and *Ruminococcus* (OTU62), norank_f_*Ruminococcaceae* (OTU853), norank_f_*Lachnospiraceae* (OTU304, 347), *Lachnospiraceae*_UCG-001 (OTU65), *unclassified_f_Lachnospiraceae* (OTU366), *Blautia* (OTU291), *Anaerotruncus* (OTU116), *norank_f_Oscillospiraceae* (OTU837), *Oscillibacter* (OTU756), *Butyricicoccus* (OTU128), *Exiguobacterium* (OTU195), and *norank_f_Desulfovibrionaceae* (OTU494) were closely related to alcoholic liver injury. It has previously been reported that ganoderic acid A supplementation significantly regulated 46 key gut microbial phylotypes (OTUs) in HFD-fed mice, which were significantly associated with at least one lipid metabolism parameter based on the network-based correlation analysis [47]. According to our previous study, the key microbial phylotypes responding to *Grifola frondosa* polysaccharides (GFP) intervention were strongly associated with glucose and lipid metabolic disorder-related parameters [48].

### 3.7. Effects of MP Intervention on Liver Metabolome in Mice with Excessive Alcohol Intake

There were significant differences in the liver metabolites between the ALD model mice and healthy mice in the PCA and PLS-DA score plots (Figure 8A,B), indicating that over-drinking obviously affected liver metabolism. Furthermore, the OPLS-DA score plot revealed clear segregation between the MP and model groups (Figure 8C), suggesting that MP intervention could significantly alter liver metabolic disturbances in mice with over-drinking. The OPLS-DA loadings S-plot (Figure 8D) showed the differences in liver metabonomic profile between the MP and model groups. To investigate the effect of MP intervention on hepatic metabolism in ALD mice, a total of 75 potential biomarkers were successfully identified (*p* < 0.05 and VIP value > 1.0) by comparing the liver differential metabolites between MP-treated group and model group (Figure 8E), of which 55 liver metabolites (including LPE(0:0/18:0), LPE(0:0/20:4(5Z,8Z,11Z,14Z)), LPE(22:6(4Z,7Z,10Z,13Z,16Z,19Z)/0:0), cholesterol, acetoacetic acid, 1H-pyrrole-2-carboxaldehyde, nervonyl carnitine, etc.) were significantly down-regulated by MP intervention, and 20 liver metabolites (including PC(18:4(6Z,9Z,12Z,15Z)/18:2(9Z,12Z), 4-oxo-retinoic acid, 5-HETE, oxonantenine, styrene, etc.) were significantly increased by MP intervention.

It was previously reported that 4-oxo-retinoic acid was particularly important in increasing catalase activity, which is a major mechanism of defense against cytotoxic effects of H_2_O_2_ [49]. More recently, 4-oxo-retinoic acid has been shown to improve obesity and glucose homeostasis in vivo [50]. 5-HEPE is a major metabolite produced by the 5-lipoxygenase enzyme catalyzing from eicosapentaenoic acid (EPA). EPA was catalyzed by 5-lipoxygenase to generate 5-HEPE. EPA and 5-HEPE exert immunomodulatory effects by enhancing the induction of adipose tissue macrophages (ATM) and regulatory T cells (Tregs). Furthermore, it has been reported that 5-HETE works as an Nrf-2 activator through the metabolite 5-oxo-ETE [51]. Oxonantenine, belongs to the class of organic compounds known as aporphines, which is an Indonesian phytochemical that may computationally become a candidate for DPP-4 inhibitor (a new diabetic drug for patients with type 2 diabetes who do not achieve normal blood glucose levels using standard drugs) [52]. Styrene and its metabolite styrene oxide can be converted to glutathione by the glutathione S-transferase, and its concentrations have a significant positive correlation with the level of anti-oxidation in the liver [53]. The results suggest that the activation of the Keap1-Nrf-2 system in hepatocytes of MP, enhances the expression of antioxidant enzymes, inhibits liver cell injury, and prevents ALD.

Among various phospholipids, phosphatidylcholine (PC) and lysophosphatidylethanolamine (LPE) are important constituents of plasma membranes. Alcohol exposure was found to reduce hepatic PC and PC/PE ratios in ALD model animals [54]. LPE is an enzymatic hydrolysis product obtained by using phospholipase A1 to hydrolyze phosphatidylethanolamine (PE) after losing one molecule of fatty acid. MP supplementation decrease hepatic LPE (0:0/18:0), LPE (0:0/20:4 (5Z, 8Z, 11Z, 14Z)) and LPE (22:6(4Z,7Z,10Z,13Z,16Z,19Z)/0:0), but increases hepatic PC (18:4(6Z,9Z,12Z,15Z)/18:2(9Z,12Z)) and PC/PE ratios in overdrinking mice. In addition, MP downregulated the hepatic cholesterol. The cholesterol and LPE were also downregulated among metabolomic biomarkers of the Mongolian medicine *Polygonatum sibiricum* [55,56]. With decreased availability of carbohydrates (such as alcoholism, starvation, and diabetes), the body breaks down fat for energy instead. As well, acetyl-CoA is diverted to produce more acetoacetate, from which acetone and beta-hydroxybutyrate are formed. High levels of ketones (acetone, acetoacetate, and beta-hydroxybutyrate) in the blood become more acidic. This is known as ketoacidosis [57]. MP downregulated the content of acetoacetic acid, and inhibits the toxicity of alcohol-induced ketone body accumulation. The downregulation of nervonyl carnitine and 1H-pyrrole-2-carboxaldehyde undermine antioxidant defense by suppressing Nrf-2-ARE pathways [58]. MP attenuates the ethanol-induced perturbations in phospholipid metabolism and oxidative damage is likely to hinder the development of ALD.

To gain a complete picture of the changes in liver metabolome induced by MP intervention, metabolic pathway enrichment based on KEGG analysis was performed by MetaboAnalyst 5.0. The metabolic pathways significantly changed by MP treatment mainly included synthesis and degradation of ketone bodies, purine metabolism, histidine metabolism, butyrate metabolism, steroid biosynthesis, primary bile acid biosynthesis, steroid hormone biosynthesis, and glutathione metabolism (Figure 8F). It was previously reported that abnormal primary bile acid biosynthesis might in HFD-induced NAFLD mice lead to steatohepatitis [49].

### 3.8. Effects of MP Intervention on the mRNA Transcription Levels in Liver

The expression levels of genes related to lipid metabolism, oxidative stress, and inflammation were detected by RT-qPCR to further elucidate the mechanism of MP in preventing liver injury (Figure 9). Compared with the healthy mice, the mRNA transcription levels of fatty acid synthase (*FASn*), a cluster of differentiation 36 (*CD36*), and interleukin 6 (*IL6*) were significantly up-regulated in mice of the model group (*p* < 0.01). On the contrary, the mRNA levels of *LDLr*, cholesterol 7A hydroxylase 1 (*CYP7A1*), acyl-CoA oxidase 1 (*ACOX1*), carnitine palmitoyltransferase-1 (*CPT-1*), fatty acid transporter 5 (*FATP5*), bile salt export pump (*BSEP*), apolipoprotein E (*ApoE*), alcohol dehydrogenase 2 (*ADH2*), acetaldehyde dehydrogenase 2 (*ALDH2*), *Nrf-2*, *NQO1*, *HO-1*, *GSH-Px*, *SOD1*, *CAT*, *GSH* and interleukin 10 (*IL10*) were distinctly down-regulated in alcohol-treated mice (*p* < 0.05). Excessive drinking can promote the synthesis of free fatty acids, cholesterol, and triglycerides, while inhibiting the oxidative decomposition of lipids, leading to lipid accumulation and fatty degeneration in the liver [29]. Activation of the rate-limiting enzyme *CYP7A1* in hepatic cholesterol metabolism can significantly promote the conversion of cholesterol into bile acids, thereby reducing the level of cholesterol in the liver and blood [59]. As previously reported, the elevation of *LDLr* reduces gene expression of *HMGCR* and inhibits cholesterol synthesis, enhances the activity of acyl-coenzyme A (CoA): cholesterol acyltransferases (ACATs), and reduces free cholesterol [60]. Therefore, the activation of *LDLr* is an efficient approach to reducing serum LDL-c levels [61]. *FASn* was responsible for free hepatic fatty acids synthesis, and *CD36* increased the absorption of free fatty acids [62]. On the contrary, *ACOX1* catalyzed the first step of peroxisome β-oxidation, and *CPT-1* was a key regulatory and rate-limiting enzyme in the β-oxidation of long-chain fatty acids [63]. MP intervention thereby reduced liver damage caused by the accumulation of fatty acids in the liver [64,65]. Apolipoprotein E (*ApoE)* participates in the synthesis, secretion, processing, and metabolism of lipoproteins. *ApoE* is a ligand of *LDLr* and a ligand of the chylomicron (CM) remnant receptor of hepatocytes, which promotes the metabolism of LDL, VLDL, and HDL [66]. *BSEP* is a key transporter of bile acid efflux, mainly transporting glycine and taurine-conjugated bile salt to complete the enterohepatic circulation [67]. The nuclear transcription factor *Nrf-2* undergoes nuclear translocation, combines with the antioxidant response element (ARE), and activates a large number downstream the transcription of antioxidant enzyme genes (*NQO1*, *HO-1*, *CAT*, *SOD1*, *GSH-Px,* and *GSH*), play a protective role against oxidation. Our study found that MP may act as a novel *Nrf-2* activator, thus regulating gene expression of lipid metabolism and antioxidant in the liver. Moreover, high-dose MP intervention also enhanced the expression levels of alcohol decomposition genes (*ADH2* and *ALDH2*) to prevent alcohol-induced liver damage. It is well known that *ADH2* and *ALDH2* are key rate-limiting enzymes of EtOH metabolism in the liver. At first, *ADH2* decomposes EtOH to acetaldehyde, and then *ALDH2* catalyzes the oxidation of acetaldehyde to acetic acid [67]. As is known to all, the abnormal expression of *ADH2* and *ALDH2* genes in the liver is extremely related to the occurrence of various liver diseases [66]. *IL-6*, an alcohol-induced inflammatory response-related gene, is closely related to the occurrence of some diseases such as pulmonary fibrosis, breast neoplasms, and postmenopausal osteoporosis [68]. *IL-10*, an anti-inflammatory cytokine, protects sensitized mice against hepatic injury induced by lipopolysaccharide or staphylococcal enterotoxin B [69]. Interestingly, our results found that MP intervention remarkably restrained the mRNA transcription of *FASn*, *CD36,* and *IL6*, but increased the mRNA transcription of *LDLr*, *CYP7A1*, *ACOX1*, *CPT-1*, *ApoE*, *FATP5*, *BSEP*, *ADH2*, *ALDH2*, *Nrf-2*, *NQO1*, *HO-1*, *GSH-Px*, *SOD1, CAT, GSH* and *IL10* (*p* < 0.05).

### 3.9. Effects of MP Intervention on the Protein Expression in Liver

Compared with the control group, the EtOH-treated mice had significantly higher levels of *SREBP-1* protein expression, but significantly lower levels of *LDLr*, *CPT-1*, *Nrf-2*, *NQO1,* and *HO-1* protein expression. However, MP intervention distinctly increased the protein expression levels of *LDLr*, *CPT-1*, *Nrf-2*, *NQO1,* and *HO-1* in alcohol-treated mice (Figure 10A,B). On the contrary, the protein expression level of *SREBP-1*, a major transcription factor that regulates the expression of genes (including *LDLr* and *HMGCR*) involved in cholesterol, fatty acid, and triglyceride biosynthesis [70], was significantly down-regulated by MP intervention. Most of the cholesterols are bound to LDL-C, which is metabolized and cleared by *LDLr* on the surface of liver cells. Therefore, up-regulation of *LDLr* protein expression is one of the most important ways to reduce blood cholesterol concentration [71]. As a key regulator and rate-limiting enzyme of fatty acid β-oxidation, *CPT-1* plays an important role in the transport of fatty acids across the inner mitochondrial membrane [72]. Previously, MP intervention effectively activated the *pAMPK* and *PPAR-γ* (upstream protein of *CPT-1*) and promoted the oxidation of liver fatty acid [7]. In this study, high-dose MP intervention alleviated liver injury through the *Nrf-2/HO-1/NQO1* signaling pathway. It can also be seen from the previous liver biochemical analysis that MP intervention significantly increased the activities of hepatic antioxidant enzymes and significantly decreased the level of hepatic lipid peroxidation. These findings may explain the significant reduction in triglyceride and lipid peroxidation levels in the liver of mice with ALD under MP intervention.

## 4. Conclusions

The protective effects of MP from RMR on liver injury and its underlying mechanism were firstly explored from the perspective of intestinal microbiomics and liver metabolomics, combined with liver gene transcription and protein expression analysis. 16S rRNA amplicon sequencing demonstrated that MP intervention improved intestinal bacterial dysbiosis by altering the proportion of some intestinal bacterial taxa in mice with excessive drinking. Liver metabolomics analysis showed that MP intervention regulated taurine and hypotaurine metabolism, riboflavin metabolism, primary bile acid biosynthesis, etc. Results of RT-qPCR and WB revealed that MP intervention regulated mRNA transcription and protein expression of the genes involved in lipid metabolism, antioxidative status, and inflammatory response in the liver. To conclude, these findings revealed that dietary MP intervention repairs alcoholic liver injury partly through the regulation of the gut microbiome and liver metabolic pathway, providing useful information for excavating functional foods to prevent ALD. In future research, the beneficial effect of MP intervention on alcoholic liver injury needs to be verified through clinical population trials.

## Figures and Tables

**Figure 1 foods-11-03048-f001:**
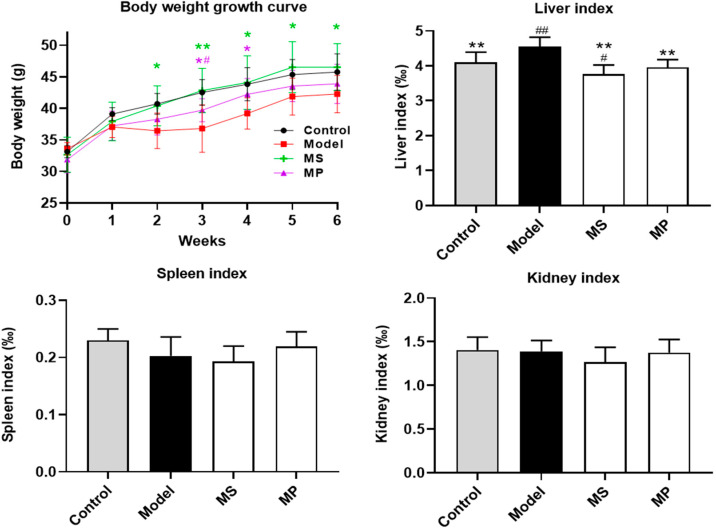
Effects of MP intervention on the body weight and organ index in mice with excessive drinking. Data with different superscripts are significantly different from each other at *p* < 0.05. ^##^
*p* < 0.01 and ^#^ *p* < 0.05, versus the Control group; ** *p* < 0.01 and * *p* < 0.05, versus the Model group.

**Figure 2 foods-11-03048-f002:**
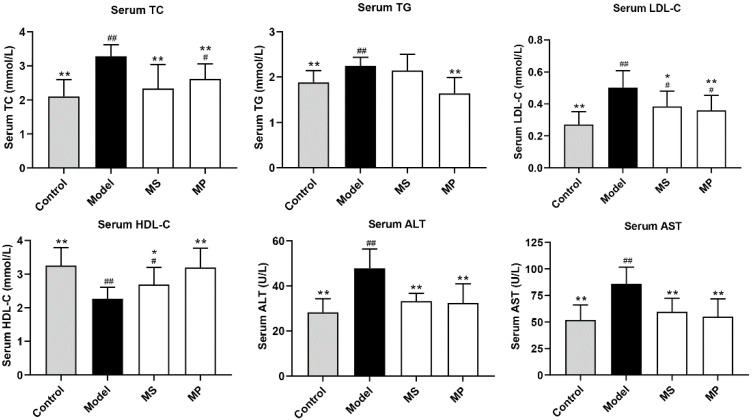
Effects of MP on serum biochemical parameters in mice with excessive drinking. Data with different superscripts are significantly different from each other at *p* < 0.05. ^##^
*p* < 0.01 and ^#^ *p* < 0.05, versus the Control group; ** *p* < 0.01 and * *p* < 0.05, versus the Model group.

**Figure 3 foods-11-03048-f003:**
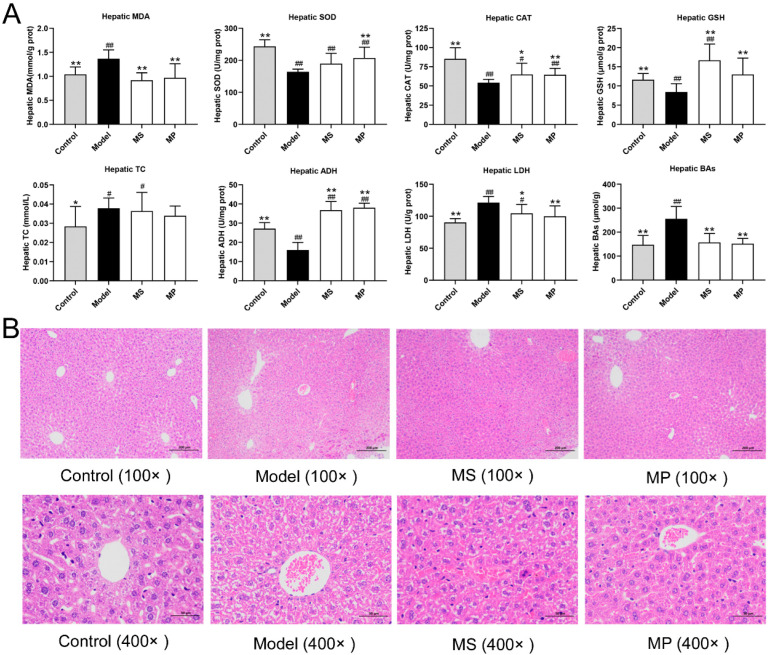
Effects of MP intervention on the liver biochemical parameters and histopathological features in mice. (**A**) Hepatic levels of MDA, SOD, CAT, GSH, TC, ADH, LDH, and BAs. (**B**) Histological changes in the liver. Data with different superscripts are significantly different from each other at *p* < 0.05. ^##^
*p* < 0.01 and ^#^ *p* < 0.05, versus the Control group; ** *p* < 0.01 and * *p* < 0.05, versus the Model group.

**Figure 4 foods-11-03048-f004:**
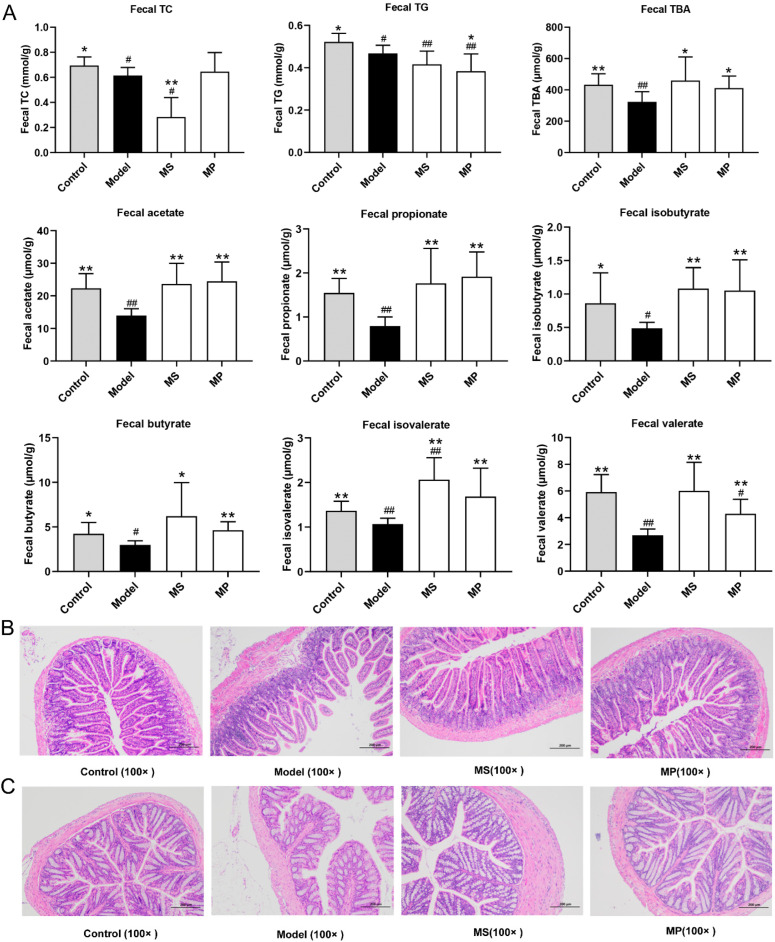
Effects of MP intervention on the fecal biochemical parameters and intestinal histopathological features in mice. (**A**) Fecal levels of TC, TG, TBA, and fecal SCFAs. (**B**) Histopathological features of jejunum. (**C**) Histopathological features of the colon. Data with different superscripts are significantly different from each other at *p* < 0.05. ^##^
*p* < 0.01 and ^#^ *p* < 0.05, versus the Control group; ** *p* < 0.01 and * *p* < 0.05, versus the Model group.

**Figure 5 foods-11-03048-f005:**
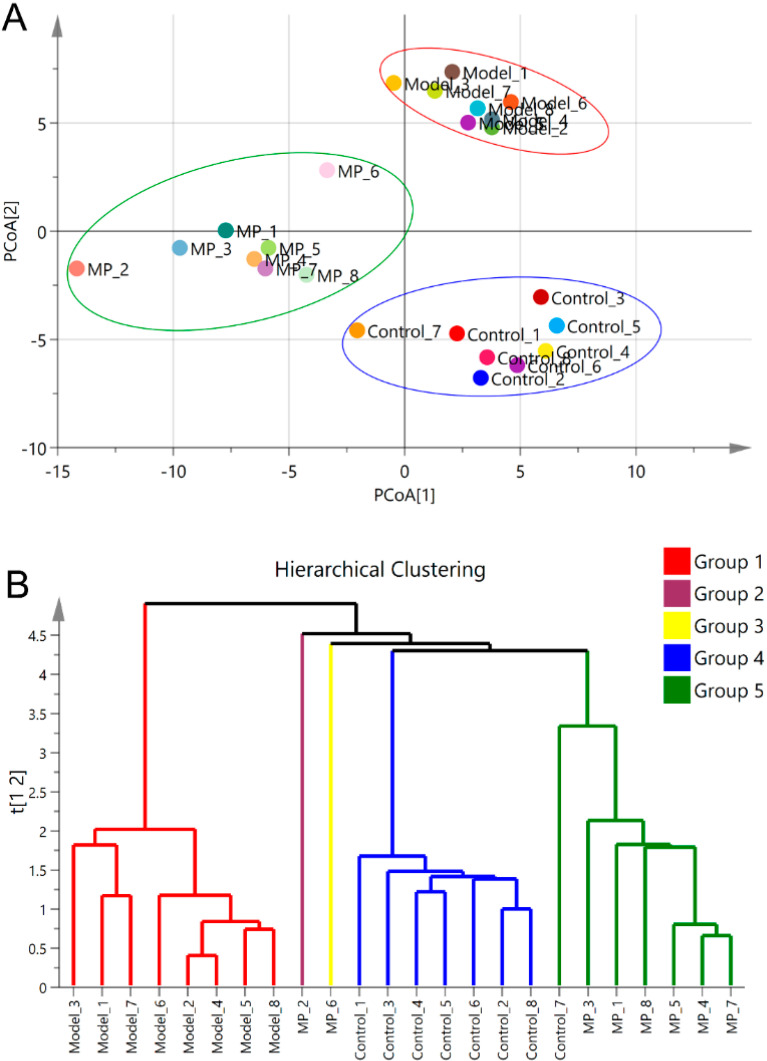
Effects of MP intervention on the composition of intestinal microflora in ALD mice. (**A**) Score plot of principal coordinate analysis (PCoA). (**B**) Hierarchical clustering plot of the intestinal microbiota composition.

**Figure 6 foods-11-03048-f006:**
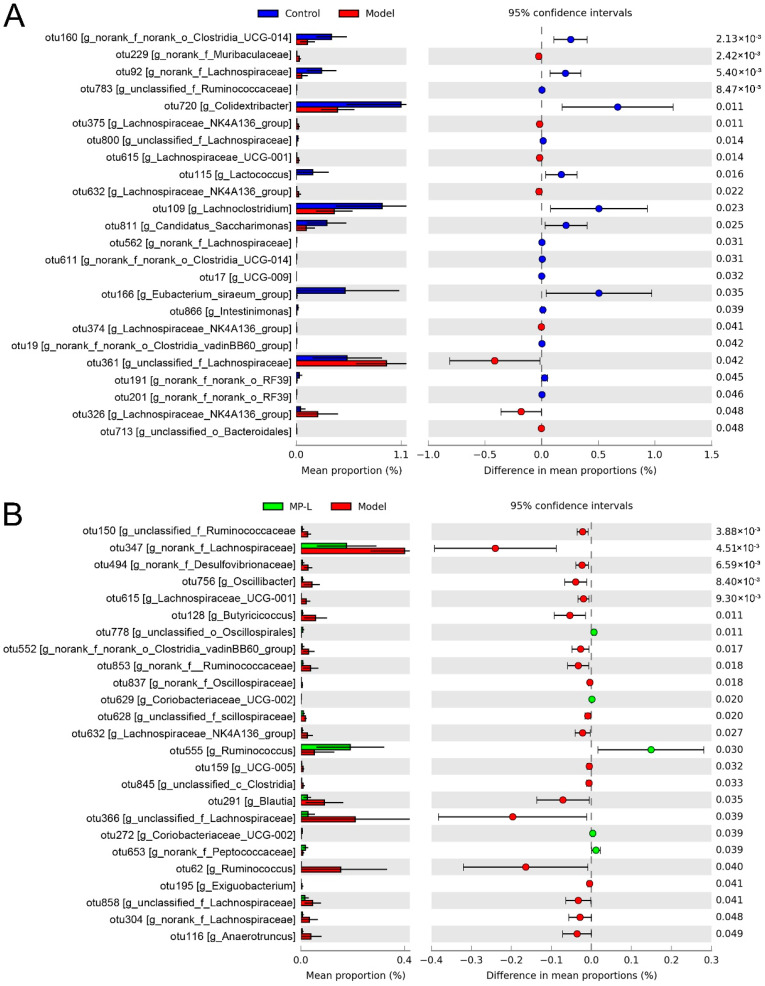
Differences in the abundance of intestinal bacterial genera between two groups. (**A**) The control group vs. the model groups. (**B**) The MP group vs. the model groups.

**Figure 7 foods-11-03048-f007:**
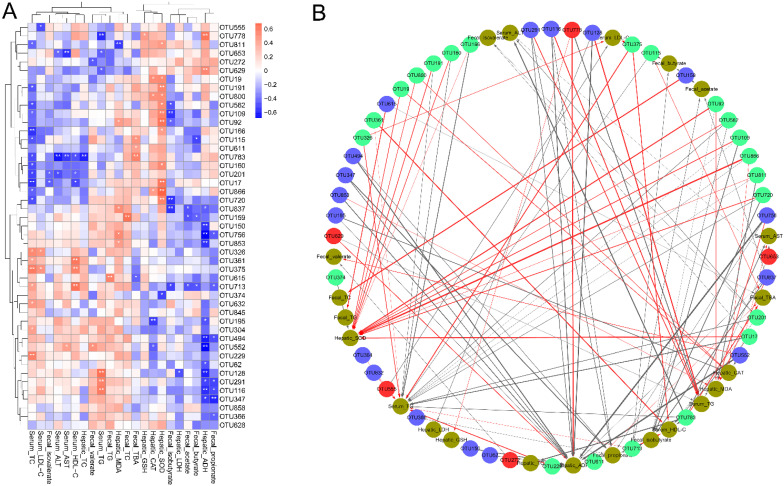
Spearman’s correlations between the key intestinal bacterial taxa and biochemical parameters. (**A**) Heatmap of correlation coefficients between the key intestinal bacterial taxa and biochemical parameters. Note: The blue square is a negative correlation, the red square is a positive correlation, and ** *p* < 0.01 and * *p* < 0.05 indicate extremely significant correlation or significant correlation, respectively. (**B**) Visualization of the significant correlation network according to Spearman’s correlations between the key intestinal bacterial taxa and biochemical parameters (|r| > 0.6, FDR adjusted *p* < 0.01). Note: Each node represents intestinal bacterial taxon and biochemical parameters. Red/blue/green nodes: the key intestinal microbial OTUs were significantly up-regulated/down-regulated/not regulated by MP, respectively; yellow nodes: biochemical parameters. The red line and the black line represent positive and negative correlations, respectively. In addition, the line width indicates the strength of correlation.

**Figure 8 foods-11-03048-f008:**
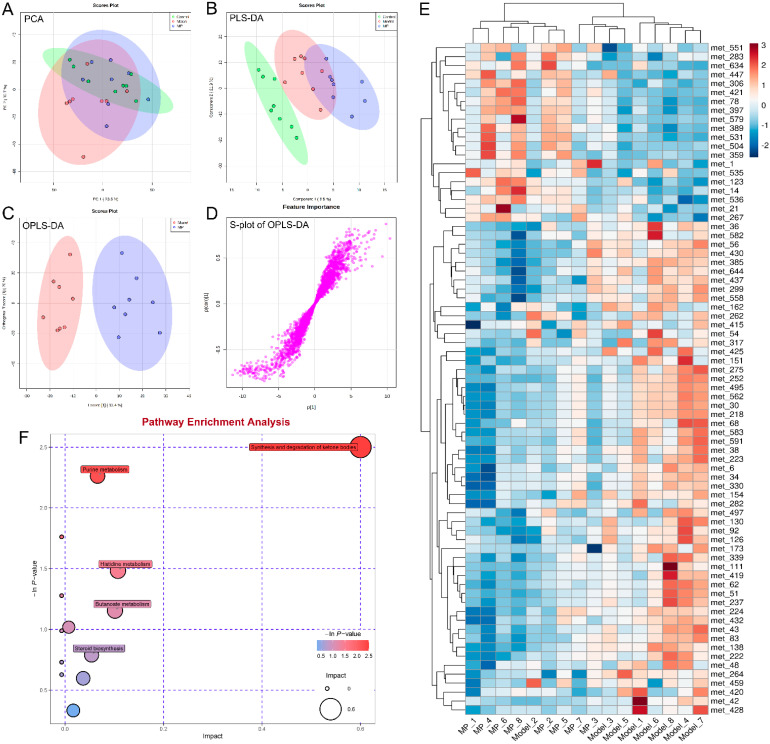
Liver metabolomic analyzed with UPLC-QTOF MS. (**A**) PCA score plot; (**B**) PLS-DA score plot; (**C**) OPLS-DA score plot; (**D**) S-loading plot according to the OPLS-DA analysis; (**E**) Heatmap of the relative abundance of significantly differential metabolites (VIP value > 1.0, *p* < 0.05) between the Model group and MP group; (**F**) Metabolic pathway analysis.

**Figure 9 foods-11-03048-f009:**
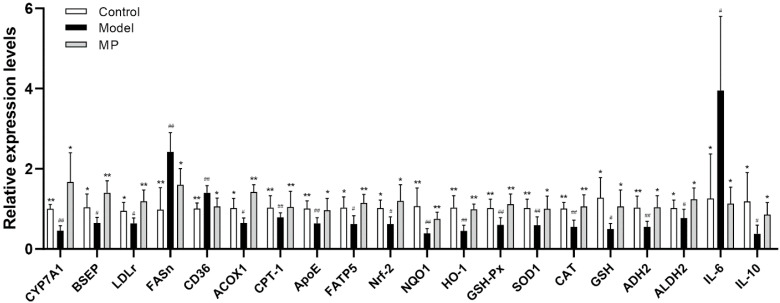
Effects of MP intervention on the mRNA levels of genes responsible for lipid metabolism, oxidative stress, and inflammatory response in livers of ALD mice. Data with different superscripts are significantly different from each other at *p* < 0.05. ^##^
*p* < 0.01 and ^#^ *p* < 0.05, versus the Control group; ** *p* < 0.01 and * *p* < 0.05, versus the Model group.

**Figure 10 foods-11-03048-f010:**
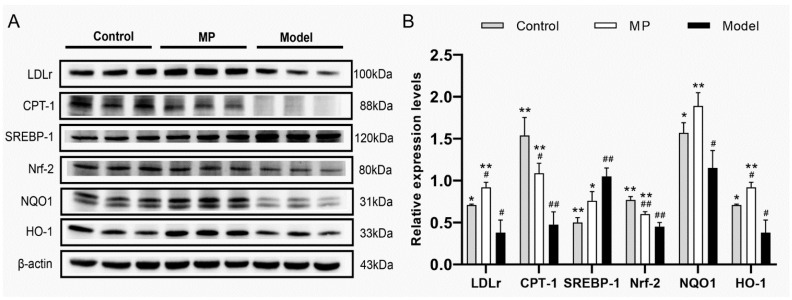
Effects of MP intervention on the protein expression levels in the liver of ALD mice. (**A**) Immunoblotting photograph. (**B**) The relative expression levels normalized against β-actin. Data with different superscripts are significantly different from each other at *p* < 0.05. ^##^
*p* < 0.01 and ^#^ *p* < 0.05, versus the Control group; ** *p* < 0.01 and * *p* < 0.05, versus the Model group.

**Table 1 foods-11-03048-t001:** Primer sequences for RT-qPCR.

Gene	Forward Primer (5′−3′)	Reverse Primer (5′−3′)
*LDLr*	ATGCTGGAGATAGAGTGGAGTT	CCGCCAAGATCAAAKAG
*CYP7A1*	CCTTGGGACGTTTTCCTGCT	GCGCTCTTTGATTTAGAKG
*SREBP-1c*	GCCGGCGCCATGGACGAGCTGG	CAGAKGGCTTCCAGAGAGGAG
*FASn*	CTGCCACAACTCTGAGGACA	TTCGTACCTCCTTGGCAAAC
*CD36*	ACTTGGGATTGGAGTGGTGATGT	GGATACCTGCAGTTTGAGCCA
*ACOX1*	GCCTGCTGTGTGGGTATGTCATT	GTCATGGGCGGGTGCAT
*CPT-1*	TCCATGCATACCAAAGTGGA	TGGTAGGAGAGCAGCACCTT
*ApoE*	AKCCGCTTCTGGGATTACCT	TCAGTGCCGTCAGTTCTTGTG
*FATP5*	AKTCGGGAGGCAGAAKCT	AGCGGGTCATACAAGTGAGC
*ADH2*	AACGGTGAAKGTTCCCAAAA	ACGACCCCCAGCCTAATACA
*ALDH2*	ATCCTCGGCTACATCAAATCG	GTCTTTTACGTCCCCGAACAC
*Nrf-2*	CCGGGAKCAAKCAGAKA	ACGTTGTCCCCATTTTTGCG
*NQO1*	AKCCCAGTCTATGCCCCAC	GGCGTGCAAGGGATGATTTC
*HO-1*	AACAAGCAAKCCCAGTCTATGC	AGGTAGCGGGTATATGCGTGGGCC
*GSH-Px*	GGGACCCTGAGACTTAGAGC	AATCCGTACTAGCGCTCACA
*SOD1*	TTGGCCGTACAA GGTGG	CGCAATCCCAATCACTCCAC
*CAT*	TCACCCACGATATCACCAGA	AGCTGAGCCTGACTCTCCAG
*GSH*	ACCAGGAAKTGGCAAAK	TCCTCCTTTCTTCCCACCG
*IL-6*	TTCTCTGGGAAATCGTGGAAA	TGCAAGTGCATCATCGTTGT
*IL-10*	CAG AGCCACATGCTCCTAGA	GCTTGGCAACCCAAGTAA CC
*BSEP*	TCTGACTCAGTGATTCTTCGCA	CCCATAAACATCAGCCAGTTGT
Mouse 18S	AGTCCCTGCCCTTTGTACACA	CGATCCCAGGGCCTCACTA

## Data Availability

Data is contained within the article or Appendix A.

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
