# Peer review of "Monascuspiloin from Monascus-Fermented Red Mold Rice Alleviates Alcoholic Liver Injury and Modulates Intestinal Microbiota"

_foods, 2022, doi:10.3390/foods11193048_

Round 1

Reviewer 1 Report

The paper showed that administration of MP improves the condition of alcoholic liver injury from the perspective of intestinal microbiomics and liver metabolomics. The steps and methods are good organization. The results and conclusion are good written. I put a small comment on this manuscript:  

1. Please add more references in the introduction section to show that novelty of this research.

2. Please mention in method or discussion why the author uses MS or MP at 10 mg/kg b.w./day?

3. Same as no 1, please add more reference in the result and discussion section to show more evidence of these results. 

4. Why did the author not use MS at all experiments? After section 3.3 (in result and discussion section) the author only uses MP.     

Author Response

Reviewer 1

The paper showed that administration of MP improves the condition of alcoholic liver injury from the perspective of intestinal microbiomics and liver metabolomics. The steps and methods are good organization. The results and conclusion are good written. I put a small comment on this manuscript:  

Q1. Please add more references in the introduction section to show the novelty of this research.

Response: Thanks for the reviewer’s comment. It is really true as the reviewer pointed out that we should add more references in the Introduction section to show the novelty of this research. We have revised the Introduction section of the manuscript accordingly. “Although a large number of therapeutic drugs have been developed for the treatment of alcoholic liver injury, some of them have certain side effects. The drug disulfiram, approved by the Food and Drug Administration (FDA) of the United States in 1983 to treat alcoholic liver disease, has little evidence that it enhances abstinence, is poorly tolerated and is now rarely used. Naltrexone, approved in 1995 to treat alcoholism, controls alcohol cravings but can also cause liver cell damage [3].” has been added in the Introduction section. “According to untargeted liver metabolomics analysis, the FXR/RXR activation pathway, which is closely related to bile acid synthesis, was most significantly up-regulated in ALD animals [17]. Alcohol-regulated liver metabolite levels were restored after treatment with ginger essential oil [18]. Therefore, liver metabolomics analysis is helpful to comprehensively understand the potential pathway of MYP for the prevention of ALD and elucidate the molecular mechanism of its hepatoprotective effect.” has been added in the Introduction section.

References:

[3]. O’Shea, R.S.; Dasarathy, S.; McCullough, A.J.; Alcoholic Liver Disease. Official journal of the American College of Gastroenterology | ACG 2010; 105:14-32.

[17]. Georgia, C.; Tan, W.Y.; PabloOrlicky, B.C.; David J., O.; Jaya Prakash, G.; Rolando, G.M.; et al. Liver metabolomics identifies bile acid profile changes at early stages of alcoholic liver disease in mice. Chemico-Biological Interactions 2022; 360:109931.

[18]. Liu, C.T,; Raghu, R.; Lin, S.H.; Wang, S.Y.; Kuo, C.H.; Tseng, Y.J.; et al. Metabolomics of ginger essential oil against alcoholic fatty liver in mice. J Agric Food Chem 2013; 61(46):11231-40.

Q2. Please mention in method or discussion why the author uses MS or MP at 10 mg/kg b.w./day?

Response: Thanks for the reviewer’s comment. We are very sorry for our negligence in the description of the daily consumption of monascuspiloin (MP) and monascin (MS). It is really true as the reviewer pointed out that we should describe the principles for setting the dosage of MS and MP used in this study. The dose settings of monascuspiloin (MP) and monascin (MS) in this study were mainly based on the daily consumption of Monascus-fermented red mold rice recommended by the Chinese Nutrition Society and relevant literature reports [7,20-23]. It has previously been reported that MS (3.75~15.0 mg/kg b.w./day) has hepatoprotective, anti-inflammatory, and hypoglycemic effects in mice [7,20-23]. “Based on the experimental dosage of MS in many previous studies on alcohol-induced injury [7, 20-21], anti-inflammatory [22], and hypoglycemic [23], the MS dose in our trial was determined to be 10 mg/kg b.w./day. Taking MS as a positive control, the protective effect of MP on the alcoholic liver injury was investigated at the same dosage of 10 mg/kg b.w./day to conduct a comparative study.” has been added in the Materials and Methods section- 2.2. Animal experiment.

References:

[7]. Lai, J.R.; Hsu, Y.W.; Pan, T.M.; Lee, C.L. Monascin and ankaflavin of Monascus purpureus prevent alcoholic liver disease through regulating AMPK-mediated lipid metabolism and enhancing both anti-inflammatory and anti-oxidative systems. Molecules 2021; 26(20): 6301.

[20]. Cheng, C.F.; Pan, T.M.; Protective effect of Monascus-fermented red mold rice against alcoholic liver disease by attenuating oxidative stress and inflammatory response. J Agric Food Chem 2011; 59(18): 9950–9957.

[21]. Lee, C.L.; Wen, J.Y.; Hsu, Y.W.; Pan, T.M.; Monascus-fermented yellow pigments monascin and ankaflavin, showed antiobesity effect via the suppression of differentiation and lipogenesis in obese rats fed a high-fat diet. J Agric Food Chem 2013; 61: 1493–1500.

[22]. Hsu, L.C.; Liang, Y.H.; Hsu, Y.W.; Kuo, Y.H.; Pan, T.M.; Anti-inflammatory properties of yellow and orange pigments from Monascus purpureus NTU 568. J Agric Food Chem 2013; 61: 2796–2802.

[23]. Lee, B.H.; Hsu, W.H.; Huang, T.; Chang, Y.Y.; Hsu, Y.W.; Pan, T.M.; Monascin improves diabetes and dyslipidemia by regulating PPARγ and inhibiting lipogenesis in fructose-rich diet-induced C57BL/6 mice. Food & Func 2013; 4: 950–959.

Q3. Same as no 1, please add more reference in the result and discussion section to show more evidence of these results. 

Response: Thanks for the reviewer’s comment. We have revised the manuscript accordingly. “It has previously been reported that ganoderic acid A supplementation significantly regulated 46 key gut microbial phylotypes (OTUs) in HFD-fed mice, which were significantly associated with at least one lipid metabolism parameter based on the network-based correlation analysis [47]. According to our previous study, the key microbial phylotypes responding to Grifola frondosa polysaccharides (GFP) intervention were strongly associated with glucose and lipid metabolic disorder related parameters [48].” has been added in the Result and Discussion section-3.6. Correlations of the key bacterial OTUs with the biochemical parameters

References:

[47]. Guo, W. L.; Guo, J. B.; Liu, B. Y.; Lu, J. Q.; Chen, M.; Liu, B.; et al. Ganoderic acid A from Ganoderma lucidum ameliorates lipid metabolism and alters gut microbiota composition in hyperlipidemic mice fed a high-fat diet. Food & Func 2020;11(8): 6818–6833.

[48]. Guo, W. L.; Deng, J.C.; Pan, Y.Y.; Xu, J.X.; Hong, J.L.; Shi, F.F.; et al. Hypoglycemic and hypolipidemic activities of grifola frondosa polysaccharides and their relationships with the modulation of intestinal microflora in diabetic mice induced by high-fat diet and streptozotocin. Int J Biol Macromol 2020; 153:1231–1240.

 Q4. Why did the author not use MS at all experiments? After section 3.3 (in result and discussion section) the author only uses MP.     

Response: Thanks for the reviewer’s comment. We are very sorry for our negligence in the description of why we did not use MS in all experiments. In fact, it has previously been reported that MS prevents alcoholic liver disease through modulating AMPK-mediated lipid metabolism and enhancing both anti-inflammatory and anti-oxidative systems [7]. In other words, the mechanism by which MS prevents alcoholic liver injury has been relatively clear. In our study, MS was only used as a positive control in animal experiments. First of all, by comparative analysis of relevant biochemical parameters in Sections 3.1-3.4, we confirmed that MP intervention does have a significant protective effect on alcohol-induced liver injury. Then, in Sections 3.4-3.9, we focused on the prevention mechanism of MP against alcohol-induced liver injury.

References:

[7]. Lai, J.R.; Hsu, Y.W.; Pan, T.M.; Lee, C.L. Monascin and ankaflavin of Monascus purpureus prevent alcoholic liver disease through regulating AMPK-mediated lipid metabolism and enhancing both anti-inflammatory and anti-oxidative systems. Molecules 2021; 26(20): 6301.

[17]. Georgia, C.; Tan, W.Y.; PabloOrlicky, B.C.; David J., O.; Jaya Prakash, G.; Rolando, G.M.;et al. Liver metabolomics identifies bile acid profile changes at early stages of alcoholic liver disease in mice. Chemico-Biological Interactions 2022; 360: 109931.

[18]. Liu, C.T,; Raghu, R.; Lin, S.H.; Wang, S.Y.; Kuo, C.H.; Tseng, Y.J.; et al. Metabolomics of ginger essential oil against alcoholic fatty liver in mice. J Agric Food Chem 2013; 61(46): 11231-40.

Reviewer 2 Report

The article examines the influence of monascuspiloin from fermented rice on alcoholic liver injury in mice. Made an impressive complex study of monascuspiloin action with the methods of metabolomics, genomics, and patho-histology. I have no objections to the authors, the text is beautifully written, and the results are illustrated with sufficient evidence.

However, I believe that in metagenomic research it is good to give a complete profile of the species present in the sample. Indeed, the authors only focused on the species that were affected by excessive drinking, but even so, since they have data on the complete metagenome, it would be better to present it. Also, I think there is no need to write the OTU when presenting the bacterial species in the text, as this information is present in two of the figures below and only makes it difficult to read. It would be good if the metagenome is deposited in NCBI, please, give the accession number.

Author Response

Reviewer 2

The article examines the influence of monascuspiloin from fermented rice on alcoholic liver injury in mice. Made an impressive complex study of monascuspiloin action with the methods of metabolomics, genomics, and pathohistology. I have no objections to the authors, the text is beautifully written, and the results are illustrated with sufficient evidence. However, I believe that in metagenomic research it is good to give a complete profile of the species present in the sample. Indeed, the authors only focused on the species that were affected by excessive drinking, but even so, since they have data on the complete metagenome, it would be better to present it. Also, I think there is no need to write the OTU when presenting the bacterial species in the text, as this information is present in two of the figures below and only makes it difficult to read. It would be good if the metagenome is deposited in NCBI, please, give the accession number.

Response: Thanks for the reviewer’s comment. We have revised the manuscript accordingly. The sequence data reported in this study was archived in the Sequence Read Archive (SRA) with the accession number PRJNA872915. (http://www.ncbi.nlm.nih.gov/bioproject/872915). It was added in the method section 2.6.